# Development of AGT-7: An Innovative ^99m^Tc-Labeled Theranostic Platform for Glioblastoma Imaging and Therapy

**DOI:** 10.3390/ph18081175

**Published:** 2025-08-08

**Authors:** Stavroula G. Kyrkou, Vasileios-Panagiotis Bistas, Evangelia-Alexandra Salvanou, Timothy Crook, Maria Giannakopoulou, Vasiliki Zoi, Maximos Leonardos, Andreas Fotopoulos, Chrissa Sioka, Ioannis Leonardos, George A. Alexiou, Penelope Bouziotis, Andreas G. Tzakos

**Affiliations:** 1Department of Chemistry, Section of Organic Chemistry and Biochemistry, University of Ioannina, 45110 Ioannina, Greece; st.kyrkou@uoi.gr (S.G.K.); bilbistas@gmail.com (V.-P.B.); 2Institute of Nuclear & Radiological Sciences & Technology, Energy & Safety, National Center for Scientific Research “Demokritos”, 15341 Athens, Greece; Salvanou@rrp.demokritos.gr; 3John Fulcher Neuro-Oncology Laboratory, Department of Brain Sciences, Division of Neuroscience, Faculty of Medicine, Imperial College London, London W12 0NN, UK; tr.crook@gmail.com; 4Neurosurgical Institute, School of Medicine, University of Ioannina, 45500 Ioannina, Greece; m.giannakopoulou91@gmail.com (M.G.); vasozoi95@gmail.com (V.Z.); andreas.fotopoulos9@gmail.com (A.F.); csioka@yahoo.com (C.S.); galexiou@uoi.gr (G.A.A.); 5Laboratory of Zoology, Department of Biological Applications and Technologies, University of Ioannina, 45110 Ioannina, Greece; m.leonardos@uoi.gr (M.L.); ileonard@uoi.gr (I.L.)

**Keywords:** glioblastoma, Tetrofosmin, temozolamide, theranostics, technetium-99m molecular imaging, SPECT, radiopharmaceuticals

## Abstract

**Background**: Glioblastoma, the most common malignant primary brain tumor in adults, continues to present a major therapeutic challenge, with a median survival of only 12–15 months and a 5-year survival rate below 2%. Despite aggressive treatment—including maximal surgical excision, radiation, and temozolomide (TMZ) chemotherapy—recurrent disease is nearly universal due to the tumor’s infiltrative nature. **Objectives:** To address the critical need for improved diagnostic and therapeutic strategies for glioblastoma multiforme (GBM), we have developed an innovative theranostic molecule, [^99m^Tc]Tc-**AGT-7**. **Methods**: **AGT-7** integrates diagnostic and therapeutic modalities comprising [^99m^Tc]Tc-TF (a nuclear medicine imaging agent) and TMZ. The diagnostic component has been tailored to selectively accumulate in glioma mitochondria. A chelating moiety enables radiolabeling with technetium-99m (^99m^Tc) for precise Single-Photon Emission Computed Tomography (SPECT) imaging. The therapeutic arm includes the tethering of a TMZ moiety for localized cytotoxicity. **Conclusions**: In vitro studies illustrated that **AGT-7** has potent cytotoxic effects in GBM cell lines (T98 and U87), with greater efficacy than TMZ, and toxicity assays in zebrafish embryos indicated a favorable safety profile. Biodistribution studies in CFW mice demonstrated that [^99m^Tc]Tc-**AGT-7** exhibited a ~10-fold lower heart uptake compared to [^99m^Tc]Tc-TF, implying reduced off-target cardiac localization. This significantly lowers the risk of cardiotoxicity and enhances **AGT-7**’s potential as a glioma-targeted theranostic agent.

## 1. Introduction

Glioblastoma, accounting for approximately 50% of adult malignant brain tumors, remains one of the most lethal CNS (Central Nervous System) diseases [1,2]. Despite its relatively low incidence rate (approximately 5/100,000), the aggressive phenotype of this cancer is associated with a median survival of 15 months after initial diagnosis [3,4]. Compounds targeted to treat glioblastoma multiforme (GBM) are limited in efficacy due to its infiltrative growth, rapid proliferation, and intrinsic tumor heterogeneity. Additionally, distinct interactions with the brain microenvironment and the presence of the blood–brain barrier (BBB) represent additional challenges to the design of effective anti-GBM agents [5]. The BBB complicates the development of treatments for intracranial malignancies by limiting access to only lipophilic and low molecular weight (less than 400 Da) drugs [6]. It is well-recognized that surgical resection alone leads to a median survival of about 5 months. Radiotherapy alone can increase survival to 10–12 months, whereas the addition of the alkylating agent temozolomide (TMZ), in addition to radiation, increases survival to a median of 15 months [7,8]. Currently, the initial phase of management involves surgical excision, when possible, followed by a combination of radiotherapy and concomitant and adjuvant chemotherapy [9]. TMZ is the most commonly utilized chemotherapy drug, often employed as a primary therapeutic option. The United States Food and Drug Administration (USFDA) approved TMZ, also recognized under the brand name Temodal, in 2005 as a second-generation DNA alkylating agent [10]. This medication is generally well-tolerated and exhibits effective penetration of the BBB, which is a major determinant of its therapeutic efficacy [11]. Specific clinical protocols inform TMZ’s dose density and intensity during concurrent radiation therapy, which has an average duration of six months [8]. Furthermore, studies have demonstrated that administering TMZ over a longer time period has some beneficial effects [12]. Recognized toxicities of TMZ include myelosuppression and nausea. However, the most important clinical factor limiting its efficacy is innate and acquired resistance to activity [8]. Understanding the mechanism of drug resistance is pivotal, and a closer examination of TMZ’s mechanisms of action sheds light on this phenomenon.

TMZ is a prodrug and therefore initially inert [10]. Upon hydrolysis, it is converted to the active metabolite methyltriazen-1-yl imidazole-4-carboxamide (MTIC), activating the production of the reactive DNA methylating species methyl hydrazine. Of note, this hydrolysis occurs at a normal pH (pH > 7) without requiring liver metabolism. Subsequently, genomic DNA undergoes methylation at specific sites: N7 (>70%), O6 (6%) of guanine, and N3 (9%) of adenine. Although methylation at the O6 position of guanine (O6-MeG) occurs rarely, it is highly significant, because it can cause base-pairing errors during DNA replication, ultimately leading to cytotoxicity. This comprehensive understanding of the mechanism of TMZ’s action provides insights into the intricate dynamics involved in drug resistance [11,13]. In GBM cells, robust DNA repair pathways effectively rectify TMZ-mediated O6-methylguanine adducts, ultimately leading to chemoresistance. Three primary DNA repair pathways contribute to TMZ resistance: Mismatch Repair (MMR), O6-methylguanine-DNA methyltransferase (MGMT), and Base Excision Repair (BER) via the poly (ADP)-ribose polymerase (PARP) pathway. Of note, MGMT critically counteracts TMZ-induced cytotoxicity through the direct repair of methylation-derived O6-methylguanine (O6-MeG) adducts. Understanding these repair mechanisms is essential in unraveling the intricate dynamics of TMZ resistance in GBM [13]. Extensive research efforts have been dedicated to addressing the challenges associated with GBM and its resistance to systemic anti-cancer therapy, such as with TMZ. These efforts demonstrate a commitment to advancing understanding of the intricate mechanisms operating and an aim to develop strategies to overcome the formidable obstacles posed by this aggressive form of cancer [14]. The significant progress in research shows a shared commitment to improving treatments, and ultimately it will improve outcomes for individuals facing the complex challenges of GBM [3,15,16].

Although TMZ remains a key agent in GBM management, the requirement for precise imaging agents to monitor treatment response and disease progression has driven interest in radiopharmaceuticals [17]. A well-known radio-probe that informed the present study and is widely used for diagnostic purposes is Technetium-99m Tetrofosmin ([^99m^Tc]Tc-TF). [^99m^Tc]Tc-TF, also known by its brand name Myoview^TM^, which was introduced in 1993 by Kelley et al. [18]. These investigators sought to develop a novel class of radiopharmaceuticals intended for use in myocardial perfusion imaging (MPI). Since then, [^99m^Tc]-TF has been the standard of excellence for myocardial perfusion imaging in clinical practice. Because it is safe, cost-effective, and produces high-quality images, it is widely employed [19]. Researchers have also used [^99m^Tc]-TF in (Single-Photon Emission Computed Tomography/Computed Tomography SPECT/CT) imaging investigations, including Stacy et al. [20]. It was observed that [^99m^Tc]-TF helps to detect reduced blood circulation in cases of lower body vascular blockage, particularly in regions such as the calf muscles. This diagnostic utility is useful for evaluating diseases such as peripheral arterial disease.

Tetrofosmin (TF) is a compound that acts as a ligand that can complex ^99m^Tc. The molecule formed is a positively charged complex that functions as a hybrid of TF and technetium-99m (^99m^Tc). The term “lipophilic di-oxo-monocation complex” refers to the formation of a central technetium atom utilizing two identical di-phosphine ligands [21]. The uptake of ^99m^Tc-TF is closely associated with energy-dependent transport pathways, cellular ion homeostasis, and mitochondrial membrane potentials due to its positive charge and lipophilic nature [19].

^99m^Tc-TF has been used for brain tumor imaging. It can enter viable cells through passive transport due to its positively charged chemical nature. In previous studies, ^99m^Tc-TF was shown to help distinguish between glioma recurrence and necrosis induced by radio chemotherapy, differentiate low- from high-grade tumor, and assess response to treatment and overall prognosis in glioblastoma patients [22,23]. Furthermore, [^99m^Tc]Tc-TF uptake in glioblastoma cells has been correlated with the sensitivity of the cells to TMZ. Together, these properties imply that ^99m^Tc-TF may be a useful tool for assessing the early response to treatment in patients with high-grade gliomas [24]. Due to its positive charge of the complex, uptake is strongly related to both the cell membrane and mitochondrial potentials. Furthermore, ^99m^Tc-TF proved superior to ^99m^Tc-MIBI for brain tumor imaging. Its uptake is less dependent on the multidrug resistance phenotype of glioblastoma, both in vitro and in vivo [20552283, 25436147]. In a study by Arbab et al., ^99m^Tc-TF accumulated inside the mitochondria of HBL-2 cells to only a low extent, in comparison to ^99m^Tc-MIBI [25].

The ability of [^99m^Tc]Tc-TF to accumulate in mitochondria can also be exploited for targeted delivery to this specific organelle. By preserving the molecular features that facilitate mitochondrial localization, such as its positive charge, while retaining the radiometal component to serve diagnostic purposes, it is possible to design a novel compound with these combined properties. This compound could also serve as a potential guide for therapeutic strategies targeting subcellular organelles. This characteristic aligns with the growing interest in exploiting mitochondrial dysfunction, particularly in cancers where mitochondria are overexpressed. Mitochondrial dysfunction is a key characteristic of neoplasia [26]. Due to substantial changes in the mitochondrial genome, gliomas can impair mitochondrial function, resulting in an altered morphology and unusual bioenergetics [26]. Hence, targeting mitochondria with photosensitizers has gained the interest of several scientific groups. However, a strong issue that remains is the premature leakage of photosensitizers before reaching mitochondria. With the development of nanocarriers, this issue has been at least partially resolved. Recently, brain endothelial cell-derived extracellular vesicles were used to encapsulate a positively charged lipophilic molecule that targets mitochondria and a photosensitizer. This complex effectively inhibited GBM cell growth in vivo, without causing systemic toxicity [27]. 

Beyond mitochondrial targeting, as previously mentioned, another noteworthy characteristic of [^99m^Tc]Tc-TF is its diagnostic potential through nuclear imaging. For SPECT or Positron Emission Tomography (PET) imaging applications, short-lived gamma- or positron-emitting radionuclides such as ^99m^Tc, gallium-68 (^68^Ga), iodine-123 (^123^I), indium-111 (^111^In), fluorine-18 (^18^F), and copper-64 (^64^Cu), are extensively used for radiolabeling various molecules. Due to excellent nuclear characteristics such as the emission of low-energy gamma rays (140 keV) and a short half-life (t_1/2_ = 6.02 h), as well as its convenient availability from ^99^Mo/^99m^Tc generators, ^99m^Tc is the most commonly used radionuclide in diagnostic imaging [28]. A very efficient way of radiolabeling compounds with ^99m^Tc was first introduced by Alberto et al. in 1998 and involves the [^99m^Tc][Tc(H_2_O)_3_(CO)_3_]^+^ carbonyl core. This method enables short radiolabeling procedures without compound modification with bifunctional chelators, under mild conditions, and it has been widely used in the development of new radiopharmaceuticals [29].

In the present work, we designed a compound that bears a ^99m^Tc affinity similar to that of [^99m^Tc]Tc-TF. This molecule is engineered to selectively localize within glioma cells by targeting overexpressed mitochondria. Additionally, it functions as a carrier for the cytotoxic agent TMZ, combining targeted imaging with an effective therapeutic capacity.

The overall objective of our study was the development of an innovative theranostic molecule that provides the simultaneous imaging and therapy of glioblastoma, addressing the unmet clinical need for the development of innovative compounds that combine both a diagnostic entity and a therapeutic payload, while offering reduced potential side effects compared to existing compounds within the same theranostic class [30,31,32,33]. The diagnostic component is based on a TF derivative radiolabeled with ^99m^Tc, while the payload comprises TMZ attached to the TF derivative, which also serves as the targeting agent.

## 2. Results

### 2.1. Rational Design of Technetium-99m AGT-7 ([^99m^Tc]Tc-AGT-7) as a Multifunctional Compound Bearing a Therapeutic and Diagnostic Modality

Three key pillars guided the development of the compound [^99m^Tc]Tc-**AGT-7**: imaging, targeting, and treatment (Figure 1). Our objective was to harness ^99m^Tc as an imaging agent, prompting the incorporation of a moiety specifically designed to chelate it. However, a critical aspect of our approach also involved targeting the overexpressed mitochondria in glioblastoma cells [34]. To address this, [^99m^Tc]Tc-**AGT-7** was rationally engineered to exploit this mitochondrial overabundance. This design takes advantage of the mitochondrial membrane potential (∆Ψm), which is present on both sides of the inner mitochondrial membrane (IMM) and is typically negative, ranging from 180 to 200 mV [35]. Due to this membrane potential, compounds conjugated with lipophilic cations can accumulate within mitochondria inside cells [36]. We, therefore, strategically designed the compound to bear a positive charge to enhance its glioma-targeting efficacy. Furthermore, we aimed to utilize TMZ as a therapeutic agent, given its well-established efficacy in treating patients with brain tumors [12]. The specific architecture could maximize the therapeutic potential while minimizing off-target effects on healthy cells.

### 2.2. Synthetic Route for AGT-7

The synthetic route of **AGT-7** starts with the formation of the chelating substituent through a double reductive amination with the linker molecule, followed by the removal of the BOC protecting groups, leading to the release of the secondary amine. The latter then reacts with TMZ-COOH via an amide bond, yielding the final product (Figure 1).

### 2.3. Liquid Chromatography–Mass Spectrometry (LC-MS)-Based Plasma Stability of AGT-7

**AGT-7** demonstrated a plasma stability profile very similar to that of the parent drug, TMZ. Throughout the 9-h incubation period, only minimal stability differences were observed. This slight variation is likely due to the compound being a structural analogue of TMZ, which could clarify its similar behaviour in plasma. Overall, the results suggest that the compound retains the stability profile of TMZ (Figure 2).

### 2.4. Cytotoxic Evaluation of AGT-7 in Glioma Cell Lines

To explore the effect of **AGT-7** on the viability of GBM cell lines, T98 and U87 cells were cultured with escalating concentrations of the compounds for 72 h, and afterward, the trypan blue exclusion assay was carried out. The results demonstrate a more pronounced decrease in cell viability at higher concentrations of the compound, indicating a concentration-dependent inhibitory effect on both cell lines. Specifically, the IC50 values of **AGT-7** found at 72 h post-treatment were 27 μM for the T98 cell line and 30 μM for the U87 cell line (Figure 3). In comparison, the IC_50_ values for TMZ were 143 μM in T98 cells and 50 μM in U87 cells, indicating the markedly higher potency of **AGT-7**, particularly in the TMZ-resistant T98 line (Figure 4).

### 2.5. Flow Cytometric Analysis of DNA Cell Cycle After Treatment with AGT-7

To investigate the effects of **AGT-7** on cell cycle progression in T98 and U87 cell lines, cells were treated with an IC50 value of **AGT-7** for 72 h. The results demonstrate that there was no significant effect on cell cycle progression in both cell lines (Figure 5). 

### 2.6. In Vivo Toxicity Evaluation and LD_50_ Evaluation of AGT-7 in Zebrafish Embryos

Prior to advancing the biological evaluation of **AGT-7**, it was essential to assess its in vivo acute toxicity. To this end, we employed a standard acute exposure assay using zebrafish (Danio rerio) embryos as a vertebrate model system. Drug exposure was for 96 h post-fertilization (hpf), following the guidelines established by the Organisation for Economic Co-operation and Development (OECD, 2010) for fish embryo toxicity testing. As illustrated in Figure 6, **AGT-7** exhibited a dose-dependent increase in mortality, with an LD_50_ value of 46.81 μM, and LD_25_ and LD_75_ values of 35.54 μM and 61.64 μM, respectively. Notably, as shown in Figure 6, no significant mortality was observed at concentrations up to 30 μM -a range that corresponds to the IC_50_ values observed in glioblastoma T98 (27 μM) and U87 (30 μM) cell lines. This alignment between therapeutically effective concentrations and sub-lethal doses in vivo underscores a favorable safety profile for **AGT-7** and supports its further preclinical development.

### 2.7. Radiolabeling and In Vitro Stability

Radiolabeling of [^99m^Tc]Tc-TF was achieved after the addition of freshly eluted [^99m^Tc]TcO_4_^−^ to a kit formulation of Tetrofosmin (Myoview^®^). Radiochemical purity was assessed by radio thin-layer chromatography (TLC) and was >97% (R_f_~0.6). Colloid formation was <2%, while free pertechnetate was ~1% (Figure 7).

Radiolabeling of **AGT-7** with the [^99m^Tc][Tc(CO)_3_(OH_2_)_3_]^+^ precursor was accomplished after 45 min at 40 °C. Radiochemical yields > 95% were achieved, and the radiotracer was used without further purification (Figure 8). The semi-aqua ion [^99m^Tc][Tc(CO)_3_(OH_2_)_3_]^+^ has three available coordination sites, and the metal center is primarily found in the oxidative state +1; so after radiolabeling, the water molecules were substituted with the three nitrogen atoms, thus forming the ^99m^Tc complex. The radiolabeled compound remained stable at RT for at least 24 h. [^99m^Tc]Tc-**AGT-7** was also stable in serum up to 24 h post-radiolabeling (Figure 9).

### 2.8. Lipophilicity Studies—Determination of Partition Coefficient

Lipophilicity studies were performed for the radiotracer [^99m^Tc]Tc-**AGT-7**. The lipophilicity of [^99m^Tc]Tc-TF has previously been reported but was repeated in the present study, for reasons of direct comparison. The partition coefficients (P) were determined using the shake-flask method and are expressed as logP (Table 1). While both examined compounds exhibited lipophilic behavior, higher lipophilicity values were observed for [^99m^Tc]Tc-TF, which is also reflected in the results of the biodistribution studies shown below, where the heart uptake of [^99m^Tc]Tc-**AGT-7** was significantly lower than that of [^99m^Tc]Tc-TF.

### 2.9. Ex Vivo Biodistribution Evaluation of [^99m^Tc]Tc-AGT-7 Compared to [^99m^Tc]Tc-TF

The ex vivo biodistribution of [^99m^Tc]Tc-**AGT-7** in comparison to [^99m^Tc]Tc-TF was evaluated in normal CFW mice at 1 h and 24 h post-intravenous injection via the tail vein (Figure 10, Table 2).

## 3. Discussion

We aimed to develop a theranostic compound, [^99m^Tc]Tc-**AGT-7**, with the objective of combining both diagnostic and therapeutic potential. Focusing on the therapeutic aspect, **AGT-7** demonstrated a more potent cytotoxicity profile than TMZ. We propose that the structural alteration of TMZ in **AGT-7** may have altered the original mechanism of action of TMZ. This modification could potentially enable the compound to overcome certain resistance pathways in glioblastoma cells. However, this explanation remains hypothetical and requires further mechanistic studies. Our interpretation is supported by similar findings reported in previous studies [37,38,39]. In vitro experiments further revealed that **AGT-7** does not have a significant impact on cell cycle progression in glioma cells. 

Furthermore, **AGT-7** was specifically designed to selectively target glioblastoma while minimizing undesirable off-target localization, particularly in the myocardium, which is a known site of accumulation for [^99m^Tc]Tc-TF. To verify this, we evaluated the biodistribution profile of the novel agent [^99m^Tc]Tc-**AGT-7**. We then compared it with [^99m^Tc]Tc-TF, a well-established cardiac imaging agent that, according to our previous work, also exhibits homing to glioma tissue [40]. 

As shown by our data, the biodistribution patterns of [^99m^Tc]Tc-**AGT-7** and [^99m^Tc]Tc-TF differ significantly, primarily due to their distinct physicochemical characteristics. The most notable distinction was observed in heart uptake: while [^99m^Tc]Tc-TF exhibited a very high myocardial accumulation (>20% ID/g), [^99m^Tc]Tc-**AGT-7** displayed an approximately tenfold lower uptake. This contrast reflects the higher lipophilicity of [^99m^Tc]Tc-TF, enabling its strong cardiac retention, whereas the more moderate lipophilicity/hydrophilicity balance of [^99m^Tc]Tc-**AGT-7** reduces cardiac sequestration. The reduced heart uptake of [^99m^Tc]Tc-**AGT-7** is especially desirable, as an excessive myocardial accumulation would impair its effectiveness and safety as a glioma theranostic agent. Both tracers showed elevated liver uptake at 1 h post-injection, consistent with their lipophilicity. Both compounds also exhibited a similar permeability across the ΒΒΒ. However, the slightly more hydrophilic nature of [^99m^Tc]Tc-**AGT-7** led to a higher renal clearance at 24 h post-injection, suggesting a more favorable excretion pathway. Additionally, the low stomach uptake for [^99m^Tc]Tc-**AGT-7** indicates in vivo radiochemical stability. These findings are consistent with previous literature from other research groups [41,42]. Throughout the 24-h study period, [^99m^Tc]Tc-**AGT-7** exhibited low uptake in all other evaluated organs, reinforcing its target specificity and capitalizing on its potential as a safe and effective theranostic tool for glioblastoma.

While [^99m^Tc]Tc-**AGT-7**’s cardiac uptake profile is favorable, a fraction of the compound accumulates in the liver, like the parent compound TMZ, and we are currently developing further variants of [^99m^Tc]Tc-**AGT-7** to reduce this. We are also testing novel [^99m^Tc]Tc-**AGT-7** analogues for an increased tumor homing capacity and evaluating dual magnetic resonance imaging (MRI)/fluorescent **AGT-7** analogues that could be utilized also for fluorescent-guided surgery, providing real-time intraoperative feedback.

## 4. Materials and Methods

### 4.1. Workflow Diagram

The diagram below summarizes the experimental workflow for the development and evaluation of [^99m^Tc]Tc-**AGT-7** (Figure 11). It provides an overview of the methods used in the present study, outlining each step from synthesis to biological assessment.

### 4.2. Chemicals

The chemicals and reagents utilized in this study were of analytical grade. Thermo Fisher Scientific (Karlsruhe, Germany) supplied sodium triacetoxyborohydride (Na(AcO)_3_BH) and 2-pyridinecarboxaldehyde. Fluorochem (Penrose Dock, Cork, Ireland) supplied N-Boc-1,4-butanediamine and 3-Methyl-4-oxo-3,4dihydroimidazo(5,1-d)(1,2,3,5)tetrazine-8-carboxylic acid (TMZ-COOH). 

Merck (Marousi, Athens, Greece) supplied the solvents used in the chemical synthesis, which included dichloromethane (CH_2_Cl_2_), 1,2-dichloroethane (DCE), methanol (MeOH), trifluoroacetic acid (TFA), dimethylformamide (DMF), and N,N-diisopropylethylamine (DIPEA). Avantor also supplied HPLC-grade acetonitrile (ACN). 

For column chromatography, 0.040–0.063 mm (230–400 mesh) silica gel was utilized. TLC with pre-coated Merck silica gel 60 F254 plates was used to monitor the reaction’s progress. The TLC plates were then visualized using UV light exposure. 

### 4.3. Characterization

The experiments were carried out utilizing Bruker Avance FT-NMR spectrometers (Ettlingen, Baden-Württemberg, Germany: Rudolf-Plank-Straße 23, 76275 Ettlingen): a 250 MHz instrument for recording ^1^H and ^13^C NMR spectra and a 500 MHz instrument for 1H NMR spectra and 2D ^1^H/^13^C-NMR experiments. 

For direct infusion analysis, the Xevo G2 Q-TOF mass spectrometer from Waters (Milford, MA, USA) was set to positive electrospray ionization (ESI) mode and used full-scan MS throughout a mass range of 50–1000 m/z. To maximize ion intensity, the following optimized source settings were used: capillary voltage of 3.5 kV, sample cone voltage of 120 V, source temperature of 120 °C, desolvation temperature of 250 °C, cone gas flow rate of 100 L h^−1^, and desolvation gas (N_2_) flow rate of 600 L h^−1^. These settings improved sensitivity and ionization efficiency during the analysis.

Product purity was further evaluated by an Agilent analytical HPLC (Santa Clara, CA, USA) system equipped with an InfinityLab Poroshell 120 EC-C18 column (4.6 × 150 mm) and a diode array detector (DAD). Detection was carried out at a wavelength of 254 nm. The analysis was carried out under a gradient solvent system, from A (2%) and B (98%) to A (100%) and B (0%), where solvent A consisted of ACN with 0.1% TFA, and solvent B was H_2_O with 0.1% TFA. This was followed by a 3-min phase with A (100%) and B (0%), and finally, the system reverted to the initial conditions of A (2%) and B (98%) for an additional 2 min. The flow rate was maintained at 1.0 mL/min over a 20-min period. The peak corresponding to **AGT-7** appeared at 5.15 min, indicating a purity of 95.3% (Appendix A).

### 4.4. Synthesis

#### 4.4.1. Synthesis of Tert-Butyl (4-(Bis(pyridin-2-Ylmethyl)amino)butyl)carbamate [43]

2-Pyridinecarboxyaldehyde (334 μL, 3.51 mmol), tert-butyl N-(4-aminobutyl)carbamate (305 μL, 1.59 mmol), and DCE (15 mL) were added to a two-neck round-bottom flask. The flask was purged with nitrogen gas (N_2_), and the mixture was stirred in an ice bath for 10 min. Then Na(AcO)_3_BH (845 mg, 3.98 mmol) was added, and the reaction was further stirred for 10 min in the ice bath and 3.5 h at room temperature. The reaction was quenched with 10 mL H_2_O and stirred for 1.5 h (Figure 2). The pH of the mixture was adjusted to 10 with the addition of drops of a 2.5 M sodium hydroxide (NaOH) solution and then extracted with CH_2_Cl_2_ (3 × 10 mL). The organic layers were collected, dried with sodium sulfate anhydrous (Na_2_SO_4_), and filtered. The solvent was removed under reduced pressure in a rotary evaporator. The crude mixture was purified with silica gel column chromatography, starting from 100% CH_2_Cl_2_ up to 5% MeOH/CH_2_Cl_2_ as eluent. The collected fraction containing the product was concentrated to dryness under reduced pressure in the rotary evaporator to afford a dark yellow oil (511.628 mg, yield = 86.8%). ^1^H-NMR (250 MHz, CDCl_3_-d, δ ppm): 8.51 (ddd, *J* = 4.9, 1.9, 0.9 Hz, 2H), 7.63 (td, *J* = 7.6, 1.8 Hz, 2H), 7.48 (dt, *J* = 7.9, 1.2 Hz, 2H), 7.13 (ddd, *J* = 7.5, 4.9, 1.3 Hz, 2H), 4.78 (s, 1H), 3.79 (s, 4H), 3.04 (q, *J* = 6.5 Hz, 2H), 2.54 (t, *J* = 6.9 Hz, 2H), 1.60–1.48 (m, 2H), 1.42 (s, 11H);(Appendix A).^13^C-NMR (63 MHz, DMSO-d_6_, δ ppm): 159.66, 156.14, 148.90, 136.46, 123.04, 121.98, 60.33, 53.91, 40.50, 28.42, 27.84, 24.17;(Appendix A).

#### 4.4.2. Synthesis of N1, N1-Bis(pyridin-2-Ylmethyl)butane-1,4-Diamine [43]

Tert-butyl (4-(bis(pyridin-2-ylmethyl)amino)butyl)carbamate (500 mg, 1.35 mmol) was transferred to a round-bottom flask with 10 mL of 20% TFA/CH_2_Cl_2_. The mixture was stirred at room temperature for 2 h (Figure 3). The solvent was removed under high vacuum to afford a red-brown oil. The oil was dissolved in CH_2_Cl_2_ (20 mL) and washed with H_2_O, whose pH was adjusted to 10 with the addition of drops of 2.5 M NaOH solution. The organic layer was then collected, and the aqueous layer was extracted again with CH_2_Cl_2_ (2 × 10 mL). The organic layers were collected, dried with Na_2_SO_4_, and filtered. The solvent was removed under reduced pressure in the rotary evaporator to afford an amber-colored oil (308.583 mg, yield = 84.6%). ^1^H-NMR (250 MHz, CDCl_3_-d, δ ppm): 8.54–8.44 (m, 2H), 7.62 (td, *J* = 7.6, 1.8 Hz, 2H), 7.50 (d, *J* = 7.8 Hz, 2H), 7.17–7.05 (m, 2H), 3.79 (s, 4H), 2.56 (dt, *J* = 17.2, 7.1 Hz, 4H), 1.65–1.50 (m, 2H), 1.47–1.37 (m, 2H), 1.38–1.28 (m, 2H) (Appendix A). ^13^C-NMR (63 MHz, DMSO-d_6_, δ ppm): 159.97, 148.94, 136.31, 122.83, 121.85, 60.46, 54.20, 42.02, 31.47, 24.42 (Appendix A).

#### 4.4.3. Synthesis of AGT-7

In a round-bottom flask, PyBOP (benzotriazol-1-yloxytripyrrolidinophosphonium hexafluorophosphate, 90.9 mg, 0.174 mmol), DIPEA (26 μL, 0.149 mmol), and TMZ-COOH (28.42 mg, 0.145 mmol) were added with dry DMF (2 mL). The flask was purged with N_2_. The reaction was stirred under N_2_ for 30 min at room temperature. Then N1,N1-bis(pyridin-2-ylmethyl)butane-1,4-diamine (35.8 mg, 0.132mmol) was added, and the reaction was stirred for 12 h (Figure 4). The solvent was evaporated under high vacuum. The crude mixture was purified with high-performance column chromatography, HPLC (A: H_2_O + 0.1% TFA, B: ACN + 0.1% TFA, from 70% A: 30% Β to 35% A: 65% Β 20 mL/min, 20 min, 254 nm) (Appendix A), to obtain a white solid (18.356 mg, yield = 33.4%). ^1^H NMR (500 MHz, DMSO-d_6_) δ: 8.85 (s, 1H, H-6′), 8.66–8.61 (m, 2H, H-6), 8.54 (t, *J* = 6.0 Hz, 1H, NH-13), 7.88 (td, *J* = 7.7, 1.8 Hz, 2H, H-4), 7.52 (d, *J* = 7.6, 2H, H-5), 7.44 (dd, *J* = 7.6, 5.0 Hz, 2H, H-3), 4.53 (d, *J* = 9.0 Hz, 4H, H-1), 3.88 (s, 3H, H-3′), 3.28 (q, *J* = 6.8 Hz, 2H, H-10), 3.23–3.17 (m, 2H, H-12), 1.80 (p, *J* = 7.9 Hz, 2H, H-9), 1.53 (p, *J* = 7.3 Hz, 2H, H-11) (Appendix A). ^13^C NMR (126 MHz, DMSO-d_6_) δ: 159.60 (C-14), 151.2(C-2),149.14(C-6), 139.09(C-8′), 137.45(C-4), 134.17(C-4′), 130.27(C-8′a), 128.22(C-6′), 124.53(C-3), 123.71(C-5), 53.57(C-9), 39.41(C-8′), 37.57(C-12), 35.93(C-3′), 26.08(C-11), 20.96(C-10) (Appendix A). m/z [M+H] 447,2131 (Appendix A).

### 4.5. Plasma Stability

#### 4.5.1. Sample Preparation

Plasma samples were diluted 1:1 (*v*/*v*) with phosphate buffer (pH 7.4). For each analyte, three independent samples were prepared by spiking 50 μL of a 40 μM stock solution in dimethyl sulfoxide (DMSO) into 1950 μL of diluted plasma, resulting in a final analyte concentration of 1 μM and a DMSO content of 2.5% (*v*/*v*) in plasma [44,45,46]. The mixtures were vortexed vigorously and incubated at 37 °C on a thermoshaker (Kisker Biotech, Steinfurt, Germany) at 600 rpm. Reactions were terminated at 0, 1, 2, 3, 4, 6, 9, and 24 h by transferring 150 μL of the incubated plasma into 450 μL of ice-cold ACN. After vortexing, samples were stored at −20 °C for 30 min and centrifuged, and 200 μL of the supernatant was diluted with 600 μL of water. The resulting solution was filtered through a 0.22 μm nylon syringe filter (Target Analysis, Thessaloniki, Greece) into 2 mL glass vials before analysis [47]. 

#### 4.5.2. Chromatographic Analysis

Chromatographic separation was achieved using a Waters ACQUITY i-Class Plus UPLC system (Waters, Manchester, UK) equipped with a temperature-controlled autosampler (6 °C) and a Waters ACQUITY UPLC BEH C18 column (2.1 × 50 mm, 1.7 μm particle size) maintained at 35 °C. The mobile phase consisted of (A) 0.1% formic acid in water and (B) 0.05% formic acid in ACN, delivered constantly at 0.2 mL/min. The gradient program was optimized as follows: 1% B (0.00–3.00 min), linear ramp to 100% B (3.00–10.00 min), hold at 100% B (10.00–12.00 min), return to initial conditions (12.00–12.01 min), and re-equilibration at 1% B (12.01–15.00 min). A 5 μL injection volume was employed for all analyses.

#### 4.5.3. Mass Spectrometric Detection

The UPLC system was coupled to a Xevo G2-XS QToF mass spectrometer from Waters (Milford, MA, USA) via an electrospray ionization (ESI) source operated in positive ion mode. Key ion source parameters included a capillary voltage of 1.0 kV, cone voltage of 20 V, source temperature of 120 °C, and desolvation temperature of 550 °C, with cone and desolvation gas flows set to 20 L/h and 1000 L/h (nitrogen), respectively. Mass accuracy was ensured using a LockSpray system with leucine enkephalin (5 ng/mL in 50% ACN/0.1% formic acid; 10 μL/min) as the lock mass (reference ion: [M+H]⁺ at m/z 556.2771).

#### 4.5.4. Data Acquisition and Processing

Full-scan MS data were acquired in MS^E^ mode with two alternating collision energy functions: low energy (4 eV) for precursor ions and ramped high energy (25–45 eV) for fragment ions. Scans covered m/z 100–900 at a rate of 0.3 s/scan. System control and data acquisition were managed using MassLynx v4.2 (SCN 1029), while UNIFI v1.9.4.053 facilitated data processing, including adduct identification ([M+H]⁺, [M+Na]⁺, [M+H_2_]^2+^) and relative quantitation.

### 4.6. Evaluation of Cytotoxicity in Cancer Cell Lines

The sensitivity of T98 and U87 cells to the compound was assessed using the trypan blue exclusion test. A total of 20,000 cells were seeded onto 12-well plates and incubated in a humidified atmosphere. After 24 h, cells were treated with the compound at concentrations of 20–100 μM for 72 h followed by trypan blue exclusion analysis. The results were derived as the mean of three independent experiments.

### 4.7. Flow Cytometric Analysis of DNA Cell Cycle

For the DNA cell cycle analysis, 20,000 cells were seeded in 12-well plates and after 24 h were treated with the compound at an IC50 concentration for a further 72 h. Subsequently, cells were washed with phosphate-buffered saline (PBS) solution, harvested by incubation with trypsin, and held at 37 °C for 20 min with a PI working solution (50 g/mL PI, 20 mg/mL RNase A, and 0.1% Triton X-100). PI fluorescence data were collected using a flow cytometer (Cytognos S.L., Polígono La Serna, Nave 9, 37900 Santa Marta de Tormes, Salamanca, Spain) and were analyzed using GraphPad Prism Version 6 software and MedCalc software (trial version).

### 4.8. Zebrafish Maintenance, Breeding, and Toxicity Tests

#### 4.8.1. Zebrafish Housing and Husbandry

Adult zebrafish specimens of the wild-type strain (AB) were maintained in a colony room, in a recirculated system, at 28 ± 1 °C, pH 6.5–7.5, and water conductivity of 500 ± 50 μS cm−1 with a 14 h light/10 h dark photoperiod (lights on at 8:00 a.m.). Feeding of the fish was performed twice a day with zebrafish feed (Zebrafeed, Sparos, Navi Mumbai, India) following common practices. Sexually mature zebrafish (at least three months old) were used for spawning. 

The collection of zebrafish eggs was performed at the beginning of the 14 h light phase following the mating procedure that took place overnight. After inspecting them, unfertilized eggs and those that showed developmental disorders were removed. The dechorionation process of the eggs followed at 24 h post-fertilization (hpf). 

The dechorionated embryos were placed in 24-well culture plates (2 embryos per well, 1.5 mL of solution per well), and each experiment was performed in triplicate. Preliminary tests were performed to evaluate the range of 0–100% mortality. DMSO (0.1%) was used as dissolver and as well as vehicle control (non-exposed). In the current study, seven different concentrations of **AGT-7** were tested (0.0, 10.0, 20.0, 40.0, 60.0, 80.0, and 100.0 μΜ). In total, 881 embryos were studied, of which 96 belonged to the non-exposed group. Six (6) embryos of the non-exposed group were found to be dead. Each experiment lasted 96 h post fertilization time (hpf) and began at 24 hpf, approximately at the 26-somite point, according to Kimmel et al. [48,49].

#### 4.8.2. Zebrafish Toxicity Testing

It is essential to apply an acute test that uses short-term exposure (96hpf) to **AGT-7** and then to determine the concentration that is lethal to 50% of zebrafish embryos. This is a valuable indicator of acute fish toxicity. According to OECD 2010, indications of death of an embryo include coagulation of the embryo, lack of somite formation, non-detachment of the tail, and/or lack of heartbeat.

##### Lethal Dose (LD50) Determination

Toxicity assays (LD50 values) and confidence limits (LD25 and LD75) were calculated based on cumulative mortality at the end of the experiment (96 hpf). The LD50 values were assessed using Regression Probit analysis (the chi-square test, Pearson goodness of fit test, and 95% confidence interval).

### 4.9. Radiolabeling of Myoview

Technetium-99m (^99m^Tc) is a gamma emitter with a photon energy of 140 keV, which requires radiation protection precautions during handling to reduce the risk of harm. All work associated with radiolabeling procedures was conducted in a licensed radiochemistry facility, where such experiments could be safely conducted.

All reagents and solvents had a purity of >95% and were used without further purification. ACN (>99.5%) was purchased from Carlo Erba (Val-de-Reuil, France). Dimethylsulfoxide (DMSO, >99.5%) was purchased from Aldrich Chemical (St. Louis, MO, USA). Human serum was purchased from Sigma Aldrich (St. Louis, MO, USA). Trifluoroacetic acid (>99%) was purchased from Alfa Aesar (Loughborough, UK). ^99m^Tc was eluted as Na[^99m^Tc]TcO4 from a commercial ^99^Mo^/99m^Tc generator (Mallinckrodt Medical B.V., Hazelwood, MO, USA). Commercially available [^99m^Tc]Tc-TF was used in our experiments, for reasons of comparison to the novel theranostic agent.

Analyses and separation, as well as purification processes, were performed by High-Performance Liquid Chromatography (HPLC) using a Waters 600 Controller pump, a Waters 996 Photodiode Array detector (set at 220 nm for all experiments), and a γ-RAM radioactivity detector to measure radioactive flow on a NUCLEOSIL 100-5 C18 (Macherey-Nagel, Dueren, Germany). All HPLC solvents were filtered through 0.22 mm membrane filters (Millipore, Milford, MA, USA) before use. Instant thin-layer chromatography–silica gel (ITLC-SG) 60 sheets (5 × 10 cm) (Merck, Darmstadt, Germany) were used for the determination of the radiolabeling purity of [^99m^Tc]Tc-TF (Myoview). These were developed in acetone/dichloromethane 35:65 and then measured with a Radio-TLC Scanner (Scan-Ram, LabLogic, Sheffield, UK). Radioactivity measurements were conducted in a dose calibrator (Capintec, Ramsey, NJ, USA).

The Myoview kit for radiopharmaceutical preparation contains 230 micrograms of [^99m^Tc]Tc-TF (active ingredient), 0.03 mg stannous chloride dehydrate, 0.32 mg disodium sulphosalicylate, 1.0 mg sodium D-gluconate, and 1.8 mg sodium hydrogen carbonate as a freeze-dried mixture under nitrogen. Labeling is performed by direct addition of freshly eluted sodium pertechnetate (4 mL, 20 mCi) and incubating for 15 min at room temperature. Radiolabeling yield was assessed by ITLC-SG, with a mixture of 35:65 acetone/dichloromethane as the mobile phase. In this solvent system, free [^99m^Tc]Tc-pertechnetate runs to the top piece of the strip, [^99m^Tc]Tc-TF runs to the center piece of the strip, while reduced, hydrolyzed ^99m^Tc and any hydrophilic complex impurities remain at the origin in the bottom piece of the strip. [^99m^Tc]Tc-TF should be used within 12 h of preparation.

#### 4.9.1. Preparation of the Precursor [^99m^Tc][Tc(H_2_O)_3_(CO)_3_]^+^

The labeling precursor [^99m^Tc][Tc(H_2_O)_3_(CO)_3_]^+^ was prepared as described in the literature [50]. Briefly, a vial containing 4 mg Na_2_CO_3_, 20 mg sodium tartrate, and 7 mg NaBH_4_ was sealed, and CO gas was purged for 2 min prior to the addition of 1 mL Na[^99m^Tc]TcO_4_ eluate obtained from a ^99^Mo/^99m^Tc generator. The vial was heated at 115 ◦C for 30 min and, at the end of the reaction, was left to cool at room temperature. Finally, the precursor was brought to pH 6.5–7 through the addition of Hydrochloric acid (HCl) 1 M. The formation of the precursor [^99m^Tc][Tc(H_2_O)_3_(CO)_3_]^+^ was determined by reverse-phase HPLC (RP-HPLC) on a C18-RP column by applying a linear gradient system from 0% to 100% B over 30 min at a flow rate of 1 mL/min, where solvent A was H_2_O/0.1% TFA, and solvent B was MeOH/0.1% TFA. The radioactivity of the precursor was measured using a dose calibrator. 

#### 4.9.2. Radiolabeling with [^99m^Tc][Tc(H_2_O)_3_(CO)_3_]^+^

Radiolabeling of **AGT-7** was accomplished via the ^99m^Tc carbonyl precursor (0.8–2 mCi) at 40 °C for 45 min. Radiochemical yield was assessed by HPLC, as mentioned above.

### 4.10. In Vitro Stability Studies

In vitro stability of radiolabeled **AGT-7** was evaluated at room temperature (bench stability at RT) and in the presence of serum at 37 °C up to 24 h post-radiolabeling. Serum stability was performed in order to assess the in vivo stability of [^99m^Tc]Tc-**AGT-7**. Stable binding of the radioisotope onto the novel compound is very important in order to ensure that the freely circulating radioisotope after in vivo administration will be minimal, thus avoiding excess background noise during imaging. Serum stability studies do not always ensure in vivo stability of the radiolabeled compounds under investigation; however they provide a good indication of their in vivo fate. Thus, the stability of the radiolabeled samples in the presence of human serum is tested before assessing their biological behavior in animal models.

### 4.11. Human Serum Stability

For the human serum stability assessment, 50 μL of [^99m^Tc]Tc-**AGT-7** was challenged against 450 μL human serum. The samples were incubated at 37 °C for 2 h and 24 h. Afterwards, 100 μL of each mixture was treated with 200 μL ethanol and centrifuged at 450× *g* for 10 min to precipitate serum proteins. The supernatants were removed and analyzed by HPLC, as mentioned above. The experiment was performed thrice.

### 4.12. Lipophilicity Studies

Lipophilicity studies were performed to determine the lipophilic behavior of [^99m^Tc]Tc-TF and [^99m^Tc]Tc-**AGT-7**. The lipophilicity of the radiolabeled complexes was assessed by determining the partition coefficient (P) using the shake-flask method. Briefly, in a centrifuge tube, 1 mL of 1-octanol and 1 mL PBS (0.01 M, pH 7.4) was mixed with 1–2 μCi of each of the radiolabeled complexes. The samples were vortexed for 1 min, and the radioactivity of 200 μL aliquots from each phase was measured using a gamma counter. The partition coefficient was calculated according to the following equation:(1)$$P\;=\;\frac{\frac{counts}{mL}\;on\;the\;1\;-\;octanol\;phase}{counts/mL\;on\;the\;PBS\;phase}$$

The results were expressed as logP. The procedure was repeated three times.

### 4.13. Ex Vivo Biodistribution Studies

For the ex vivo biodistribution studies, normal Carworth Farms White (CFW) mice of both genders were used. The mice were housed in individually ventilated cages (IVCs) under a constant temperature (22 ± 2 °C) and humidity (45–50%) and a 12 h light/dark cycle, with free access to food and water. The animals were obtained from the breeding facilities of the Institute of Biosciences and Applications, NCSR “Demokritos”. The experimental animal facility is registered according to Greek Presidential Decree 56/2013 (Reg. Number: EL 25 BIO 022), in accordance with European Directive 2010/63, which is harmonized with national legislation, on the protection of animals used for scientific purposes. All applicable national guidelines for the care and use of animals were followed. The study protocol was approved by the Department of Agriculture and Veterinary Service of the Prefecture of Athens (Protocol Number 634365/27-7-21). These studies have been further approved by our institutional ethics committee, and the procedures followed are in accordance with institutional guidelines. Intravenous injections were performed using insulin syringes BD Micro-Fine 1 mL (29G). The animals were euthanized in a chamber saturated with isofluorane vapors (isoflurane 1000 mg/g, Iso-Vet, Chanelle Pharma). The radioactivity of samples and syringes was measured using a dose calibrator (Capintec, Ramsey, NJ, USA), while a Cobra II automatic gamma counter (Canberra, Packard, Schwadorf, Austria), was used to measure the radioactivity of the samples from the ex vivo biodistribution studies.

The ex vivo biodistribution of [^99m^Tc]Tc-**AGT-7** was investigated at 1 and 24 h post-injection and compared to the ex vivo biodistribution of [^99m^Tc]Tc-TF. The ex vivo behavior of the two radiolabeled complexes was evaluated in mixed-gender CFW mice, 6–8 weeks old, weighing 20–30 g. The radiotracers were intravenously injected via the tail vein (100 μL/~100 μCi). At 1 and 24 h post-administration, the animals were euthanized by isoflurane inhalation, and the major organs and tissues of interest (blood, liver, heart, kidneys, stomach, intestines, spleen, muscle, lung, bone, pancreas, and brain) were collected, weighed, and measured in an automatic gamma counter. For the calculation of the injected dose in each animal, a standard solution was prepared, while the radioactivity remaining in the tail was subtracted. The syringes containing the radiolabeled samples were measured before and after injection, to determine the precise dose administered to each mouse. All measurements were corrected for background and radioactive decay. All the distribution data were calculated as the percentage of injected dose per gram (%ID/g).

## 5. Conclusions

In conclusion, we have successfully developed [^99m^Tc]Tc-**AGT-7**, a theranostic compound inspired by the glioblastoma tumor-homing properties of [^99m^Tc]Tc-TF, designed to combine both diagnostic and therapeutic functionalities. Structurally, [^99m^Tc]Tc-**AGT-7** is a modified TMZ analogue, in which the amino group of the imidazotetrazine ring is modified to tether the pharmacophore via a linker to a chelating moiety capable of binding ^99m^Tc, thereby enabling SPECT imaging potential. Cytotoxicity evaluations in glioma cell lines demonstrated significantly enhanced efficacy for **AGT-7** compared to TMZ alone, assumed to be based on the structural modification. In vivo toxicity assessments in zebrafish embryos revealed dose-dependent effects, with mortality rates increasing at concentrations higher than the IC50 values in glioma cells, indicating a favorable therapeutic window.

Radiolabeling of **AGT-7** with [^99m^Tc][Tc(CO)_3_(OH_2_)_3_]^+^ was achieved efficiently, with radiochemical yields exceeding 95% after 45 min at 40 °C. The radiolabeled compound remained stable at room temperature and in serum for at least 24 h post-labeling. Ex vivo biodistribution studies showed that [^99m^Tc]Tc-**AGT-7** can cross the blood–brain barrier in a manner comparable to [^99m^Tc]Tc-TF. Importantly, [^99m^Tc]Tc-**AGT-7** exhibited a tenfold lower cardiac uptake compared to [^99m^Tc]Tc-TF, a well-known heart imaging agent, which demonstrated an over 20% ID/g heart uptake. This reduced cardiac uptake, attributed to the lower lipophilicity of [^99m^Tc]Tc-**AGT-7**, highlights its promise as a potential glioma-targeted theranostic agent.

## Data Availability

Data is contained within the article and Appendix A.

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
