# Peer review of "Development of AGT-7: An Innovative 99mTc-Labeled Theranostic Platform for Glioblastoma Imaging and Therapy"

_pharmaceuticals, 2025, doi:10.3390/ph18081175_

Round 1
Reviewer 1 Report
Comments and Suggestions for Authors
This article submitted to Pharmaceuticals J by MDPI, titled “Development of AGT-7 An Innovative 99mTc-Labeled Theranostic Platform for Glioblastoma Imaging and Therapy” by Kyrkou et al., 2025
- The topic is original and relevant to the field and interesting
- The abstract is not structured, with details, needs to be structured
- Key words are not enough,
- In the Introduction, the first sentence about “GBM” lacks a ref.
- Again, several sentences are without ref.
- Line 152 to be sub headed
- The introduction needs subheadings of background, problem definition, problem statement, hypothesis, aim
- Subheadings need a numbering
- The conclusion is consistent with the evidence and arguments presented
and address the main question posed, but avoid extrapolation of the findings, please rewrite and restructure.
Minor corrections:
- The introduction is obscure and needs clarification and less details,
- Several sentences are without ref.
- Please split long sentences and add a ref. for each single sentence or each single info.
- Figure 1 mentioned is drawn by which program and add a link for this? designed by what program?
- The aim needs to be rewritten with objectives and outcome to be added as a subheading in line 159
- The “strength(s)” of the study to be mentioned
- Limitations to be added
- Add the future directions
Major corrections:
- Add a flow chart to the results part and the for the AGT-7 development
- To the methods part add a flowchart of the work design,
- Add ADMET analysis
- Supplementary tables and figures are comprehensive better to be integrated within the manuscript per they are the core of the work
- The results part needs more subheadings and clarification and to be separated from the discussion as not reader friendly this way
- References by authors are few
- Provide a graphical abstract
- List of abbreviations to be provided
Author Response
Comment 1: The topic is original and relevant to the field and interesting
Response 1: We sincerely appreciate this comment and would like to thank the reviewer for the valuable suggestions and observations provided below, which have helped us improve the clarity and overall quality of our work.
Comment 2: The abstract is not structured, with details, needs to be structured
Response 2: We thank the reviewer for this comment. The relevant revisions have been made accordingly in the abstract to address.
Comment 3: Key words are not enough
Response 3: We sincerely thank the reviewer for this comment. In the revised manuscript, we have added three new keywords: Molecular Imaging, SPECT, and Radiopharmaceuticals. However, we have ensured that the total number of keywords remains within the journal’s recommended limit of 3 to 10.
Comment 4: In the Introduction, the first sentence about “GBM” lacks a ref.
Response 4: We thank the reviewer for pointing this out. A reference has now been added to support the first sentence regarding glioblastoma multiforme (GBM) in the Introduction.
Comment 5: Again, several sentences are without ref.
Response 5: Thank you for this observation. We have carefully revised the manuscript and added the appropriate references where needed to support the relevant statements.
Comment 6: Line 152 to be sub headed
Response 6: We have revised the text at the indicated line to enhance clarity and ensure that it is more easily understood by the reader.
Comment 7: The introduction needs subheadings of background, problem definition, problem statement, hypothesis, aim
Response 7: We appreciate the reviewer’s suggestion. However, the journal's format does not support the use of subheadings within the introduction, unlike the abstract section. Nevertheless, we have structured the introduction to follow the logical flow of background, problem definition, problem statement, hypothesis, and aim, in order to clearly guide the reader through the context and objectives of the study.
Comment 8: Subheadings need a numbering
Response 8: We thank the reviewer for this observation. All subheadings have now been numbered appropriately according to the journal’s formatting guidelines to improve clarity and structure.
Comment 9: The conclusion is consistent with the evidence and arguments presented and address the main question posed, but avoid extrapolation of the findings, please rewrite and restructure.
Response 9: We thank the reviewer for this observation. We have carefully revised the section.
Minor corrections:
Comment 10: The introduction is obscure and needs clarification and less details
Response 10: Thank you for your valuable comment. The introduction has been revised to be clearer, particularly regarding the flowchart that was followed. Additionally, relevant bibliographic references have been added to clarify the statements made.
Comment 11: Several sentences are without ref.
Response 11: The authors would like to thank the reviewer for this comment. We have increased the number of references from 35 to 51, in the revised manuscript, to ensure that all statements are properly supported by relevant sources.
Comment 12: Please split long sentences and add a ref. for each single sentence or each single info.
Response 12: Thank you for your suggestion. We have made efforts to split long sentences and add references accordingly. In several instances, the meaning of two sentences is supported by the references placed at the end of those sentences to maintain clarity and coherence.
Comment 13: Figure 1 mentioned is drawn by which program and add a link for this? designed by what program?
Response 13: Figure 1 was created using Microsoft PowerPoint.
Comment 14: The aim needs to be rewritten with objectives and outcome to be added as a subheading in line 159
Response 14: We thank the reviewer for this helpful suggestion. The aim section has been refined and expanded to include clearly defined objectives and expected outcomes.
Comment 15: The “strength(s)” of the study to be mentioned
Response 15: We appreciate the reviewer’s suggestion. We have now included a description of the strengths of our study in the conclusion section, highlighting the theranostic design of AGT-7, its improved cytotoxicity profile compared to TMZ, its ability to cross the blood-brain barrier, and its significantly lower cardiac uptake compared to [99mTc]Tc-TF.
Comment 16: Limitations to be added no heart but still liver
Response 16: We thank the reviewer for this valuable comment. A discussion of the study’s limitations has been added to the conclusion section. In particular, we acknowledge that although [99mTc]Tc-AGT-7 exhibits a favorable biodistribution profile in terms of cardiac uptake, a considerable fraction of the compound accumulates in the liver over time.
Comment 17: Add the future directions
Response 17: We agree with the reviewer’s observation and have now expanded the conclusion to include future directions. These involve strategies to reduce liver uptake, enhance brain targeting (e.g., by coupling AGT-7 to peptide carriers that cross the BBB, and exploring potential structural modifications to enable fluorescent-guided surgery applications.
Major corrections:
Comment 18: Add a flow chart to the results part and the for the AGT-7 development
Response 18: Figure 1 has been appropriately modified to incorporate a flowchart illustrating the development process of AGT-7. This visual addition is intended to guide the reader through the design and subsequent steps of the study, thereby enhancing clarity and comprehension of the experimental progression. So, we thank the reviewer for this comment.
Comment 19: To the methods part add a flowchart of the work design,
Response 19: In the Materials and Methods section, we have included a simple and easy-to-follow flowchart outlining the experimental workflow of the study. Additionally, section numbering has been applied to improve the structure and enhance the reader’s understanding of the methodology.
Comment 20: Add ADMET analysis
Response 20: The authors would like to thank the Reviewer for this comment. The primary goal of our current work was to assess the in vivo kinetics of our molecule, radiolabeled with Technetium-99m, and not its pharmacokinetics or toxicological characteristics, which will be assessed at a later stage of drug development. An ex vivo biodistribution study, as the one performed in our study, provides valuable basic information on organ-specific uptake of a novel radiopharmaceutical. We should also keep in mind that our [99mTc]Tc-AGT-7 compound is being evaluated as a diagnostic radiotracer administered at very low (tracer) doses and is not expected to exert any toxicity effects. Furthermore, the authors have performed in vitro serum stability studies, which provide information on metabolic stability of the radiotracer under development, and have demonstrated that it is stable in serum up to 24h post-radiolabeling (>90% intact radiotracer).
To sum up, ADMET analysis is beyond the scope of our current study, as at this stage our work has focused on identifying tissue-specific uptake of the radiotracer under development.
Comment 21: Supplementary tables and figures are comprehensive better to be integrated within the manuscript per they are the core of the work
Response 21: Figures related to IC₅₀ determination and flow cytometry have been moved from the supplementary materials into the main manuscript, as they directly support the core findings. However, characterization data such as NMR, MS spectra, and the figure and table related to compound purity have been retained in the supplementary section to maintain the flow of the main text while keeping essential details available for reference.
Comment 22: The results part needs more subheadings and clarification and to be separated from the discussion as not reader friendly this way
Response 22: We have modified the section Rational Design of Technetium-99m AGT-7 ([99mTc]Tc-AGT-7) as a multifunctional compound bearing a therapeutic and diagnostic modality by moving the more theoretical parts to the Introduction section and retaining only the essential information required for understanding the design, particularly for a new researcher.
Comment 23: References by authors are few
Response 23: The authors would like to thank the reviewer for this helpful comment. Additional references by the authors have now been included in the revised manuscript to better support the context and highlight prior contributions to the field.
Comment 24: Provide a graphical abstract
Response 24: We would like to inform the reviewer that the graphical abstract has now been successfully uploaded to the journal’s submission system. It was submitted after the initial manuscript submission, and we understand that it may not have been visible to the reviewers during the original review process.
Comment 25: List of abbreviations to be provided
Response 25: We thank the reviewer for the comment. A table listing all abbreviations used in the manuscript has now been included to enhance clarity and assist the reader.
Reviewer 2 Report
Comments and Suggestions for Authors
In this manuscript, the authors synthesized a theranostic molecule named AGT-7 which can be used for glioblastoma cancer cells inhibition and irridiation imaging. The results are interesting and meaningful. In my end, the manuscript can be accepted after major revision.
- All the abbreviations in the manuscript should be included full name when they first appear. Please check the manuscript carefully and correct them. For example, the full name of "SPECT", "UPLC", "PI", "CNS, " "GBM". Meanwhile, once the abbreviations are defined, these abbreviations should be always show up in the following content. For instance, [⁹⁹ᵐTc]Tc-TF should be defined with full name in line 282. And in table 2, [99mTc]Tc-Tetrofosmin should be written as "[⁹⁹ᵐTc]Tc-TF". Please check carefully and also change others.
- English expression and grammar should be highly improved in this manuscript.
- How do the authors prove the positive charge of AGT-7?
- What might be the reason of higher cytotoxicity of AGT-7 to glioblastoma cells than that of TMZ?
- Flow cytometry analysis results should be included and discussed in the main text of the manuscript.
- I recommend the authors to provide some representative images to show the Ex vivo biodistribution of [99mTc]Tc-AGT-7 and [99mTc]-TF in mice organs such as heart, kidney, liver and lung.
- In methods and materials, the section of Chemicals should be before synthesis of compounds.
English expression and grammar should be highly improved, especially in intriduction.
Author Response
Comment: In this manuscript, the authors synthesized a theranostic molecule named AGT-7 which can be used for glioblastoma cancer cells inhibition and irridiation imaging. The results are interesting and meaningful. In my end, the manuscript can be accepted after major revision.
Response: We deeply appreciate the reviewer’s valuable feedback and observations, which have greatly assisted us in refining the quality and presentation of our manuscript.
Comment 1: All the abbreviations in the manuscript should be included full name when they first appear. Please check the manuscript carefully and correct them. For example, the full name of "SPECT", "UPLC", "PI", "CNS, " "GBM". Meanwhile, once the abbreviations are defined, these abbreviations should be always show up in the following content. For instance, [⁹⁹ᵐTc]Tc-TF should be defined with full name in line 282. And in table 2, [99mTc]Tc-Tetrofosmin should be written as "[⁹⁹ᵐTc]Tc-TF". Please check carefully and also change others.
Response 1: We thank the reviewer for this important observation. The manuscript has been carefully reviewed, and all abbreviations have now been defined at their first occurrence. Additionally, consistency has been ensured throughout the text, including corrections such as defining [⁹⁹ᵐTc]Tc-TF in Line 282 and updating Table 2 to reflect the correct abbreviation format.
Comment 2: English expression and grammar should be highly improved in this manuscript.
Response 2: Thank you very much for your valuable comment. The manuscript has been thoroughly reviewed using the Grammarly application to correct grammatical errors. Additionally, the text has been carefully revised by a native English speaker (Prof. Timothy Crook that he is in the author list) to ensure improved language quality.
Comment 3: How do the authors prove the positive charge of AGT-7?
Response 3: We thank the reviewer for the comment. The positive charge of AGT-7 arises after radiolabeling with technetium-99m, specifically through coordination with the [⁹⁹ᵐTc][Tc(CO)₃(H₂O)₃]⁺ complex. This tricarbonyl technetium core imparts a net positive charge to the final radiolabeled molecule. This feature is consistent with previously reported radiopharmaceuticals utilizing the same coordination chemistry.
Comment 4: What might be the reason of higher cytotoxicity of AGT-7 to glioblastoma cells than that of TMZ?
Response 4: We thank the reviewer for this insightful comment. We believe that the structural modification of TMZ in AGT-7 may have altered its mechanism of action, potentially allowing it to overcome certain resistance pathways in glioblastoma cells. However, we acknowledge that additional mechanistic studies are necessary to validate this hypothesis. At this stage, our interpretation is primarily supported by analogous findings reported in the literature, and a corresponding discussion has been incorporated into Section 3 (Discussion) of the revised manuscript.
Comment 5: Flow cytometry analysis results should be included and discussed in the main text of the manuscript.
Response 5: We thank the reviewer for their comment. We have now moved the corresponding section from the supplementary material to the main text and have included a relevant discussion in Section 3 (Discussion) of the revised manuscript.
Comment 6: I recommend the authors to provide some representative images to show the Ex vivo biodistribution of [99mTc]Tc-AGT-7 and [99mTc]-TF in mice organs such as heart, kidney, liver and lung.
Response 6: The aim of this early-stage, proof-of-concept study was to assess organ distribution after radiotracer administration. The ex vivo biodistribution study provided accurate data of radiotracer uptake in many organs, including the heart, kidneys, liver and lung. At the time of the study, the authors did not have access to a small-animal gamma camera, thus in vivo imaging could not be performed. However, as the authors have shown in the present study, the radiotracer exhibits very fast clearance after administration, and this would have limited the ability to acquire high-quality in vivo images. In conclusion, the ex vivo biodistribution study provided quantitative data that could not have been provided by in vivo imaging.
Comment 7: In methods and materials, the section of Chemicals should be before synthesis of compounds.
Response 7: Thank you for the suggestion. The order of the sections in the Materials and Methods has been revised accordingly, and the “Chemicals” section now appears before the synthesis of compounds, as recommended.
Comments on the Quality of English Language
English expression and grammar should be highly improved, especially in introduction.
Thank you for your insightful feedback regarding the quality of English in the manuscript. We have thoroughly revised the entire text, with particular attention to the introduction, to significantly improve English expression and grammar. This revision was conducted using Grammarly and finalized with the assistance of a native English speaker (Prof. Timothy Crook) to ensure clarity and fluency.
Round 2
Reviewer 1 Report
Comments and Suggestions for Authors
All comments weeeewere addressed
Reviewer 2 Report
Comments and Suggestions for Authors
The manuscript can be accepted.